# The Apelin receptor enhances Nodal/TGFβ signaling to ensure proper cardiac development

**Ashish R Deshwar[1,2], Serene C Chng[3], Lena Ho[3], Bruno Reversade[3,4,5]\*, Ian C Scott[1,2,6]\***

[1]Program in Developmental and Stem Cell Biology, The Hospital for Sick Children, Toronto, Canada; [2]Department of Molecular Genetics, University of Toronto, Toronto, Canada; [3]Institute of Medical Biology, A\*STAR, Singapore, Singapore; [4]Institute of Molecular and Cellular Biology, A\*STAR, Singapore, Singapore; [5]Department of Paediatrics, School of Medicine, National University of Singapore, Singapore; [6]Heart and Stroke/Richard Lewar Centre of Excellence in Cardiovascular Research, University of Toronto, Toronto, Canada

**Abstract** The Apelin receptor (Aplnr) is essential for heart development, controlling the early migration of cardiac progenitors. Here we demonstrate that in zebrafish Aplnr modulates Nodal/TGFβ signaling, a key pathway essential for mesendoderm induction and migration. Loss of Aplnr function leads to a reduction in Nodal target gene expression whereas activation of Aplnr by a non-peptide agonist increases the expression of these same targets. Furthermore, loss of Aplnr results in a delay in the expression of the cardiogenic transcription factors *mespaa/ab*. Elevating Nodal levels in *aplnra/b* morphant and double mutant embryos is sufficient to rescue cardiac differentiation defects. We demonstrate that loss of Aplnr attenuates the activity of a point source of Nodal ligands Squint and Cyclops in a non-cell autonomous manner. Our results favour a model in which Aplnr is required to fine-tune Nodal output, acting as a specific rheostat for the Nodal/TGFβ pathway during the earliest stages of cardiogenesis.

**\*For correspondence:**
bruno@reversade.com (BR);
ian.scott@sickkids.ca (ICS)

**Competing interests:** The authors declare that no competing interests exist.

## Introduction

During gastrulation, complex cell movements occur which result in the localization of progenitor populations to discrete embryonic regions for subsequent organogenesis. Loss of Apelin receptor (Aplnr) function in zebrafish, as manifested in the recessive *grinch* mutant, results in a decrease or absence of cardiogenesis, and affects expression of the earliest known cardiac mesoderm markers (*Scott et al., 2007*; *Zeng et al., 2007*). The role of Aplnr in the proper formation of the heart appears to be conserved in vertebrates. In mice, *Aplnr* (also known as *Apj* and *Agtrl1*) is expressed in the gastrulating mesoderm, with *Aplnr* mutant mice exhibiting incompletely penetrant cardiovascular malformations including thinning of the myocardium, ventricular septation defects, an enlarged right ventricle and improper heart looping (*Kang et al., 2013*). In vitro, overexpression of *Aplnr* in mouse embryonic stem cells results in enhanced cardiac differentiation of embryoid bodies, while *Aplnr* inhibition leads to impaired cardiac differentiation (*D'Aniello et al., 2013*; *D'Aniello et al., 2009*).

While a role for Aplnr signaling in the earliest events of cardiac development is evident, how Aplnr functions in this context remains unclear. In zebrafish, Aplnr has been implicated in the movement of cardiac progenitors during gastrulation to the anterior lateral plate mesoderm (ALPM), the site of heart development, with a delay in anterior migration of presumed cardiac progenitors during

**eLife digest** In one of the first events that happens as an embryo develops, cells become the different stem cell populations that form the body's organs. So what makes a cell become one stem cell type rather than another? In the case of the heart, the first important event is the activity of a signaling pathway called the Nodal/TGFβ pathway. Nodal signaling can drive cells to become many different stem cell types depending on its level of activity. Many different levels of regulation fine-tune Nodal signaling to produce these activity thresholds.

Zebrafish that have a mutation in the gene that encodes a protein called the Apelin receptor have no heart. The loss of this receptor interferes with how heart stem cells (called cardiac progenitors) are made and how they move to where heart development occurs.

Deshwar et al. have now studied mutant zebrafish in order to investigate how the Apelin receptor influences early heart development. This revealed that Nodal signaling levels are slightly lower in the mutant zebrafish embryos than in normal fish at the time when Nodal activity induces cardiac progenitors to form. When Nodal activity is experimentally boosted in zebrafish that lack the Apelin receptor, they become able to develop hearts.

Deshwar et al. also found that the Apelin receptor does not work in cells that produce or receive Nodal signals. This suggests that the Apelin receptor modulates Nodal signaling levels by acting in cells that lie between the cells that release Nodal signals and the cardiac progenitors. An important question for future work to address is how this modulation works. As Nodal is a key determinant of many cell types in developing embryos, learning how Apelin receptors regulate its activity could help researchers to derive specific cell types from cultured stem cells for use in regenerative medicine.

gastrulation (*Paskaradevan and Scott, 2012*). These early effects on gastrulation movements suggest an early function for Aplnr in cardiac development, well before expression of cardiac mesoderm genes, such as *Nkx2.5*, is initiated (*Scott et al., 2007*; *Zeng et al., 2007*; *Paskaradevan and Scott, 2012*; *Pauli et al., 2014*; *Chng et al., 2013*). Interestingly, the requirement for Aplnr in cardiac development appears to be primarily non-cell autonomous, which is to say that Aplnr is not required in the cells that will form the heart per se but rather in surrounding cells (*Paskaradevan and Scott, 2012*). The genetic deletion of *aplnrb* or its endogenous early ligand *elabela* (also known as *apela or toddler*) causes gastrulation movement defects with aberrant cardiac and endoderm development in zebrafish (*Paskaradevan and Scott, 2012*; *Pauli et al., 2014*; *Chng et al., 2013*). This, together with the ability of overexpressed Aplnr to rescue cardiac differentiation in *Cripto*-null mouse embryonic stem cells (*D'Aniello et al., 2009*), indicate a strong functional link between Aplnr and Nodal signaling for proper cardiac specification and differentiation.

In this study we report that Aplnr directly modulates Nodal/TGFβ signaling during gastrulation, a key pathway essential for mesendoderm induction and migration (*Carmany-Rampey and Schier, 2001*; *Dougan et al., 2003*). Several lines of evidence show that levels of Nodal activity are attenuated in *aplnr* mutants. Loss-of-function of Aplnr leads to a reduction in Nodal target gene expression, whereas activation of Aplnr signaling increases the expression of these same targets. By elevating Nodal levels in *aplnr* mutant/morphant embryos, we are able to restore cardiac differentiation. We find that loss of Aplnr attenuates the activity of a point source of the Nodal ligands Squint (Sqt, Ndr1) and Cyclops (Cyc, Ndr2) and that the Aplnr regulates Nodal signaling in a cell non-autonomous fashion. We propose a model in which the Aplnr fine-tunes Nodal activity during the onset of gastrulation to initiate the migration of lateral margin cells and proper heart formation. Aplnr may therefore act as a rheostat for the Nodal/TGFβ pathway.

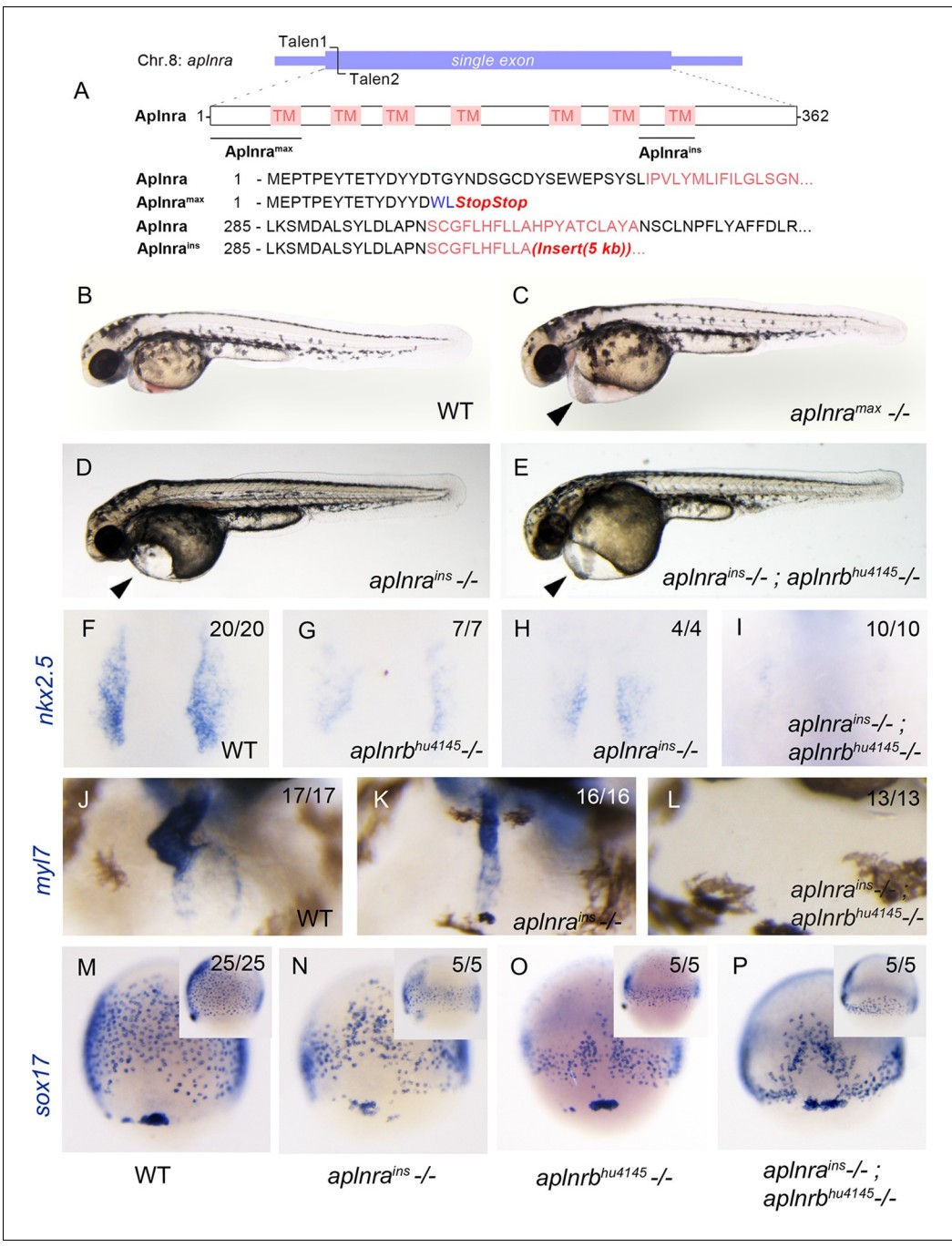

**Figure 1.** *aplnra* mutant embryos display defects in endoderm and heart formation. (**A**) Schematic detailing the *aplnra^max* and *aplnra^ins* alleles. TM indicates the transmembrane domain. (**B–E**) Gross morphology of *aplnra^max*, *aplnra^ins* and *aplnra^ins*; *aplnrb^hu4145* mutant embryos compared to WT (wild type) at 48 hpf (hours post-fertilization). (**F–I**) *nkx2.5* expression at the 15 somite stage in WT, *aplnrb^hu4145*, *aplnra^ins*, and *aplnra^ins*; *aplnrb^hu4145* mutant embryos. Dorsal view with anterior to the top. (**J–L**) In situ hybridization showing expression of *myl7* at 48 hpf in *aplnra^ins* and *aplnra^ins*; *aplnrb^hu4145* embryos compared to WT when viewed from the anterior. (**M–P**) Comparison of *sox17* expression at 8 hpf between WT, *aplnra^ins*, *aplnrb^hu4145* and *aplnra^ins*; *aplnrb^hu4145* mutant embryos. Dorsal views are shown with a lateral view in inset panels.

The following figure supplements are available for figure 1:

**Figure supplement 1.** *aplnra* and *aplnra; aplnrb* double mutant characterization.

*Figure 1 continued on next page*

*Figure 1 continued*

**Figure supplement 2.** *aplnra; aplnrb* double mutants display defects in endodermal organ development.

## Results

### Aplnra and aplnrb function redundantly in cardiac and endoderm development

The zebrafish genome harbours two paralogues (*aplnra* and *aplnrb*) of the human *APLNR* gene. Only *aplnrb*, for which the first mutant was aptly named *grinch* (*aplnrb^{s608}*, p.Trp90Leu), is known to be involved in early cardiogenesis (*Scott et al., 2007*; *Zeng et al., 2007*). In order to assess the contribution of *aplnra* to the process of gastrulation and heart development, we knocked it out using custom TALEN pairs targeted to its unique exon on chromosome 8 (*Figure 1A*). The resulting null allele, which we named *max* (the compliant dog companion of the Grinch), encodes a truncated 17-amino acid protein resulting from an early frameshift. The *aplnra^{max}* allele (p.Thr16TrpfsX2) deletes 95% of Aplnra including its seven transmembrane domains (*Figure 1A*). Present at sub-Mendelian ratios, approximately 15% of mutant larvae from heterozygous *aplnra^{max}* intercrosses showed pericardial edema (*Figure 1B–C*). As with *aplnrb* mutants (*Scott et al., 2007*; *Pauli et al., 2014*; *Chng et al., 2013*), *sox17*-positive endodermal progenitors at 8 hr post-fertilization (hpf) and *myl7*-positive cardiomyocytes at 1 day post-fertilization (dpf) were significantly reduced in numbers and intensity in a*plnra^{max}* fish (*Figure 1—figure supplement 1A–D*). Note that in this current study a novel *aplnrb^{hu4145}* (p.W54X) allele is being used. An independent allele, *aplnra^{ins}*, resulting from a viral insertion was obtained from Znomics (*Figure 1A*). Homozygous *aplnra^{ins}* embryos recapitulated the phenotype of *aplnra^{max}* and *aplnrb^{hu4145}* with similar pericardial edema (*Figure 1D*), reduced *nkx2.5*-positive cardiac mesoderm at the 15-somite stage (*Figure 1F–H*) and reduced *myl7*-positive cells at 2 dpf (*Figure 1J–K*). The number and spread of *sox17*-positive cells was significantly reduced in homozygous *aplnra^{ins}* when compared to wildtype (WT) and was not significantly different from *aplnrb^{hu4145}* single mutants (*Figure 1M–O* and *Figure 1—figure supplement 1E–F*). These *aplnra* mutant phenotypes suggest redundant functions for *aplnra* and *aplnrb*.

Double *aplnrb^{hu4145}*; *aplnra^{ins}* mutants were generated to evaluate functional redundancy for these two paralogues in early development. Double mutant embryos exhibited normal morphology at 2 dpf with pericardial edema (*Figure 1E*). In contrast to *aplnra* or *aplnrb* single mutants, which usually possess significant cardiac tissue at 2 dpf, double mutant embryos exhibited either complete absence of or an extremely small heart (*Figure 1L*). In addition, double *aplnra; aplnrb* mutant embryos exhibited a further reduction in both the spread and number of *sox17* expressing cells when compared to the *aplnra^{ins}* or *aplnrb^{hu4145}* single mutants (*Figure 1M–P* and *Figure 1—figure supplement 1E–F*). *nkx2.5* expression was negligible in double mutants suggesting a near-complete absence of early cardiac progenitors (*Figure 1I*). It should be noted that the double mutant phenotype faithfully phenocopies that seen with the injection of *aplnra/b* morpholinos (MOs) both at the morphological and molecular levels as seen by the expression of these three diagnostic markers *sox17, nkx2.5* and *myl7* (*Paskaradevan and Scott, 2012*; *Chng et al., 2013*).

Given the substantial reduction in the number of endodermal progenitors during gastrulation in *aplnra; aplnrb* double mutants, we investigated the subsequent effects on development of the endodermal-derived organs. The morphology of the gut tube was examined by performing wholemount RNA in situ hybridization (WISH) for *foxa1, foxa2* and *foxa3* at 48 hpf. While the pharyngeal endoderm appeared to be primarily intact, the most anterior population of these cells appeared to be either dramatically reduced or absent in double mutants (*Figure 1—figure supplement 2A–I*). Furthermore, the liver and pancreatic buds were consistently found to be smaller or absent, with misorientation of the pancreatic bud evident in some embryos. Taken together, this data suggests that Aplnra is required for both proper endoderm differentiation and cardiac development and that Aplnra and Aplnrb have redundant roles in these early processes.

### Aplnr activation enhances Nodal signaling during gastrulation

To gain insight into how Aplnr signaling regulates early cardiac development, we pursued a gene expression profiling approach. Comparative microarray analysis at 50% epiboly (5 hpf) of cDNA from

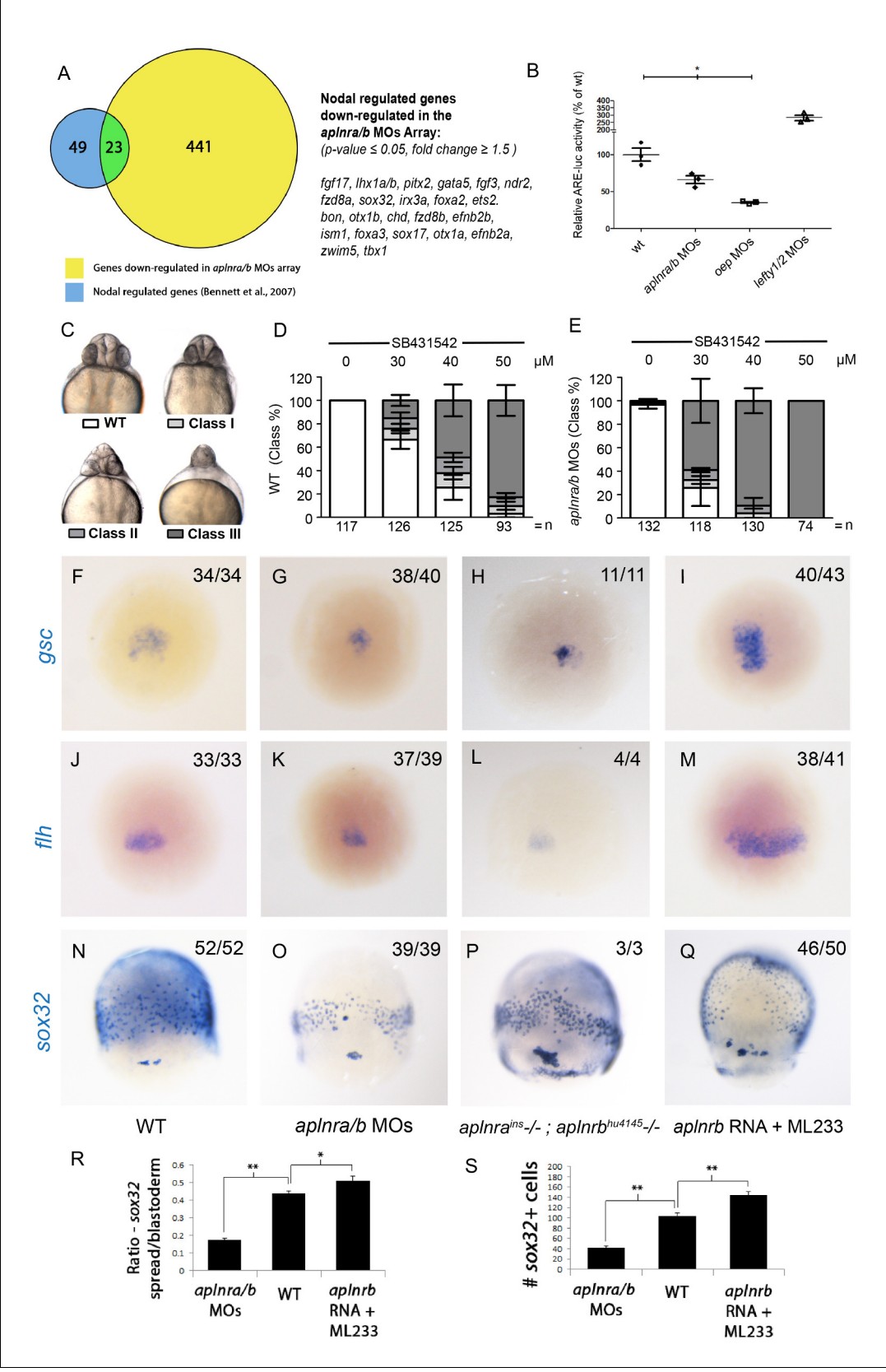

**Figure 2.** Aplnr deficient embryos exhibit a reduction in Nodal signaling. (**A**) List and Venn diagram of 23 Nodal target genes found to be down-regulated in a microarray of *aplnra/b* morphant embryos compared to WT at 50%
*Figure 2 continued on next page*

*Figure 2 continued*

epiboly (5.25 hpf). (**B**) Relative luciferase activity regulated by the Activin response element (ARE) in WT, *aplnra/b* morpholino (MO), *oep* MO and *lefty1/2* MO injected embryos at 30% epiboly (4.7 hpf). Data are represented as means ± SEM. *p<0.05 unpaired two-tailed t-test. (**C–E**) Phenotypic characterization of WT (**D**) and *aplnra/b* morphant embryos (**E**) when treated with the indicated concentration of the Alk4/5/7 inhibitor SB431542 from the sphere stage (4 hpf) onwards. (**F–S**) Visualization of the expression of the canonical nodal target genes *gsc, flh* and *sox32* in WT (**F,J,N**), *aplnra/b* MOs injected (**G,K,O**), *aplnra*[ins]; *aplnrb*[hu4145] double mutant (**H,L,P**) and *aplnrb* RNA injected treated with the Aplnr agonist ML233 (**I,M,Q**) embryos at 8 hpf. Embryos are viewed from the dorsal side. Quantification of the number and spread of *sox32* expressing cells (**R,S**). Data are represented as means ± SEM. *p<0.05, **p<0.01 unpaired two-tailed t-test.

The following figure supplements are available for figure 2:

**Figure supplement 1.** Loss of aplnr affects Nodal target gene expression.
**Figure supplement 2.** Aplnr activation enhances Nodal target gene expression.

---

WT and double *aplnra/b* morphants (injected with MOs) embryos revealed a reduction in a set of genes known to be downstream of the Nodal signaling pathway. Previous work has identified 72 Nodal-regulated genes in zebrafish at 6 hpf (*Bennett et al., 2007*). Remarkably nearly one-third (23 out of 72) of these genes were down-regulated in *aplnra/b* morphants (*Figure 2A*). The estimated probability of observing such a large overlap by chance is very small (8.6 × 10$^{-17}$ by hypergeometric distribution), suggesting that this overlap might be biologically significant, *i.e.* that Nodal signaling is decreased in the absence of Aplnr function. Conversely, using Gene Set Enrichment Analysis (GSEA) (*Subramanian et al., 2005*), we found that the genes downregulated in *aplnra/b* morphants were significantly enriched for genes upregulated in zebrafish sphere stage embryos injected with *sqt* mRNA, which encodes for one of the activating Nodal ligands Squint (*Nelson et al., 2014*), further substantiating our hypothesis that Aplnra/b promotes Nodal signaling (*Figure 2—figure supplement 1A*). Given the known role of Nodal signaling in induction and migration of the mesendoderm (*Carmany-Rampey and Schier, 2001*; *Feldman et al., 2000*; *Gritsman et al., 1999*), we surmised that Aplnr might work upstream, or in parallel, to the Nodal pathway. To more directly assess Nodal signaling levels in the embryo, we injected a Nodal/TGFβ luciferase reporter construct into WT, *aplnra/b* MOs, *oep* MO and *lefty1/2* MOs injected embryos. At 30% epiboly (4.7 hpf), embryos injected with MOs against the Nodal pathway antagonist *lefty1/2* had higher levels of Nodal/TGFβ luciferase reporter activity, while those injected with a MO against the essential Nodal co-receptor *oep* (also known as *cripto* or *tdgf1*) had lower levels (*Figure 2B*). Consistently, *aplnra/b* morphant embryos exhibited significantly lower levels of Nodal/TGFβ reporter activity compared to WT, indicating a reduction in Nodal signaling in these embryos, in agreement with our microarray results. We next sought to confirm attenuated Nodal signaling following *aplnra/b* knockdown by means of chemical inhibition. Embryos were incubated from the sphere stage (4hpf) onwards with increasing concentrations of SB431542, which acts as a dedicated Alk4/5/7 antagonist. Phenotypes were scored according to the severity of cyclopia, a hallmark feature of Nodal impairment in zebrafish (*Gritsman et al., 1999*; *Rebagliati et al., 1998*), at 2 dpf (*Figure 2C*). Following multiple independent tests (N=4) *aplnra/b* morphants were found to be significantly more sensitive to SB431542 treatment than were WT embryos (*Figure 2D–E*), suggesting reduced (but not absent) Nodal signaling levels when Aplnr is lost.

We next used WISH to assess the expression of direct downstream targets of Nodal in double *aplnra/b* morphant embryos. The canonical Nodal target genes *floating head (flh), goosecoid (gsc)* and *sox32* (*Gritsman et al., 1999*; *Chen and Schier, 2001*) all showed reduced expression in *aplnra/b* morphant embryos at 8 hpf relative to WT embryos (*Figure 2F–G, J–K, N–O*). Analysis of *sox32* expression, which marks endodermal precursors, revealed both a reduced number of endoderm cells and a decreased extent of endodermal migration (quantified in *Figure 2R–S*), consistent with previous analysis of *aplnrb* mutants (*Pauli et al., 2014*; *Chng et al., 2013*). Down-regulation of all three genes was also documented in *aplnra; aplnrb* double mutant embryos (*Figure 2H,L,P*). Further analysis of additional Nodal target genes *lefty1* and *lefty2* also revealed a decrease in expression before and at the beginning of gastrulation (*Figure 2—figure supplement 1B–G*).

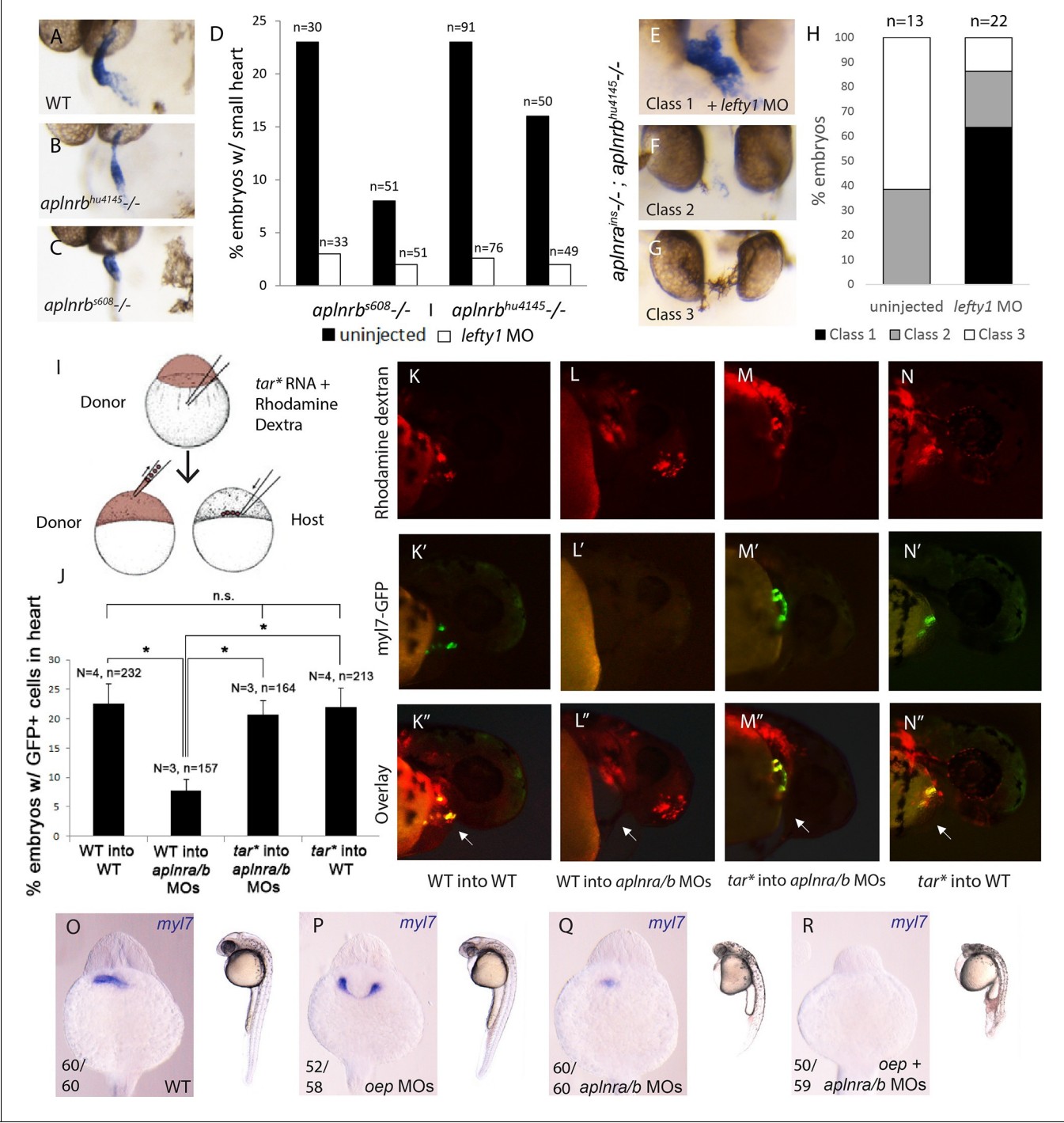

**Figure 3.** Elevation of Nodal signaling in *aplnr* mutant/morphant embryos rescues cardiogenesis. (AC) *myl7* WISH showing a representative heart phenotype at 48 hpf in a WT embryo (**A**) and two different *aplnrb* mutant alleles; *hu4145* (**B**) and *s608/grinch* (**C**). Anterior is oriented towards the left. (**D**) Quantification of the number of embryos with a small heart at 48 hpf from individual clutches of embryos in which half were injected with *lefty1* MO. Clutches were obtained from crosses of two different *aplnrb* heterozygous mutants (*hu4145* and *s608/grinch* as indicated). (**E–H**) Classification of heart phenotype in *aplnra^ins^; aplnrb^hu4145^* double mutant embryos at 48 hpf when injected with *lefty1* MO as compared to un-injected embryos. Severity of cardiac phenotypes was scored based on *myl7* WISH (**H**). (**I**) Schematic displaying the transplantation of injected donor cells into the margin of host embryos. Contribution of transplanted cells to the heart is scored based on expression of the *myl7:EGFP* transgene in donor cells. (**J–N"**) Margin transplants of WT or *tar*\* (activated Nodal receptor) overexpressing *myl7:EGFP* cells into WT or *aplnra/b* morphant embryos at 48 hpf. Arrow indicates the heart. Embryos are displayed from a lateral view with the anterior of the embryo towards the right. Data are represented as means ± SEM. \*p<0.05, n.s. = not significant, Tukey's Multiple Comparison test following significant (p<0.05) one way ANOVA. (**O—R**) Gross morphology and *myl7* expression

*Figure 3 continued on next page*

Figure 3 continued

at 24 hpf in WT (**O**), embryos injected with a sub-optimal dose of *oep* MOs (**P**), *aplnra/b* morphant embryos (**Q**) and *aplnra/b/oep* morphant embryos (**R**).

We next examined if the ectopic activation of Aplnr could be sufficient to increase the expression of Nodal target genes. Overexpression of the ligands of Aplnr, Elabela and Apelin, each result in phenotypes similar to Aplnr loss-of-function, possibly as a consequence of ligand-mediated receptor internalization and signal desensitisation (*Scott et al., 2007*; *Zeng et al., 2007*; *Paskaradevan and Scott, 2012*; *Pauli et al., 2014*; *Chng et al., 2013*). To bypass this limitation, we instead used ML233, a non-peptide small molecule agonist of Aplnr signaling (*Khan et al., 2011*). Treating embryos injected with 150 pg of *aplnrb* RNA with 2.5 µM of ML233 resulted in a significant increase in the expression of the three Nodal targets *gsc*, *flh* and *sox32* relative to WT (*Figure 2I,M,Q*) and increased both the number of endoderm cells and the extent of migration to a more anterior position (*Figure 2R–S*). Overexpression of *aplnrb,* or ML233 treatment alone, resulted in increased *flh* and *gsc* expression, whereas *sox32* expression and endoderm migration was largely unaffected (*Figure 2—figure supplement 2A–I*). ML233 had no effect on *gsc*, *flh* or *sox32* expression in *aplnra/b* morphants, indicating that the action of ML233 is Aplnr-dependent (*Figure 2—figure supplement 2J–O*). Moreover, overexpression of *aplnrb* in *oep*-depleted embryos was not sufficient to induce expression of *gsc, flh* or *sox32*, even in the presence of ML233 (*Figure 2—figure supplement 2P–U*). This argues against a scenario where Aplnr signaling is acting in parallel to Nodal signaling. Taken all together, these data suggest that Aplnr signaling is sufficient to boost endogenous levels of Nodal signaling at gastrulation stages.

## Elevated Nodal levels rescue the aplnr loss-of-function heart phenotype

As Nodal signaling is reduced in *aplnr* mutant embryos, we reasoned that increasing Nodal may ameliorate or rescue cardiogenesis in the absence of Aplnr function. To test this hypothesis, we took two complementary approaches. We first elevated the levels of endogenous Nodal signaling by injecting a MO against *lefty1*, a direct negative feedback Nodal antagonist (*Feldman et al., 2002*). *lefty1* MO was injected into embryos bearing two different *aplnrb* mutant alleles, *s608/grinch* (p. W90L) and *hu4145* (p.W54X), which exhibit a small heart (*Figure 3A–C*). While the penetrance of the heart phenotype varied within each clutch, *lefty1* MO treatment was capable of rescuing cardiogenesis in both mutants, with nearly all embryos showing rescue of the small heart phenotype (*Figure 3D*, note that 25% of embryos in a given cross would be homozygous null mutants, data for 2 independent clutches per mutant is shown). One caveat of this approach is that *aplnra/b* gene expression is regulated by Nodal signaling (*D'Aniello et al., 2009*; *Pauli et al., 2014*). It is therefore conceivable that elevating Nodal levels in *aplnrb* single mutants provides rescue simply by elevating *aplnra* gene expression. To address this issue, *lefty1* MO was injected into embryos generated from an in-cross of *aplnra; aplnrb* heterozygous parents. Embryos were evaluated for heart formation by WISH for *myl7* gene expression and subsequently genotyped. Strikingly, over 60% of *aplnra/b* double mutants exhibited proper cardiac formation when injected with *lefty1* MOs, which was not observed in un-injected mutant siblings (*Figure 3E–H*). This suggests that elevated Nodal signaling is capable of rescuing the Aplnr cardiac phenotype, even in the complete absence of Aplnr function.

As a complementary approach, we attempted to specifically elevate the levels of Nodal signaling within lateral margin cells and see if this rescued cardiac contribution. To perform this experiment, donor cells from *myl7:EGFP* transgenic embryos either injected or uninjected with *taram-a* (*tar**) RNA encoding a hyper-activated Nodal receptor (*Renucci et al., 1996*) were transplanted to the margin of *aplnra/b* morphant hosts (*Figure 3I*). As we have previously shown (*Scott et al., 2007*; *Paskaradevan and Scott, 2012*), WT cells placed at the margin of *aplnra/b* morphant hosts contributed to the myocardium at an appreciably reduced frequency (7.8%, N=3, n=157) as compared to when WT hosts were used (*Figure 3J*). In contrast, *tar** overexpressing cells, when transplanted to the margin of *aplnra/b* morphant embryos, contributed to the myocardium at a much higher frequency (20.7%, N=3, n=164) with no significant difference when compared to transplantation of WT cells into WT hosts (22.5%, N=4, n=234), suggesting a near complete rescue (*Figure 3J–N″*). It should be noted that transplantation of WT *tar** expressing cells into the margin of WT embryos did

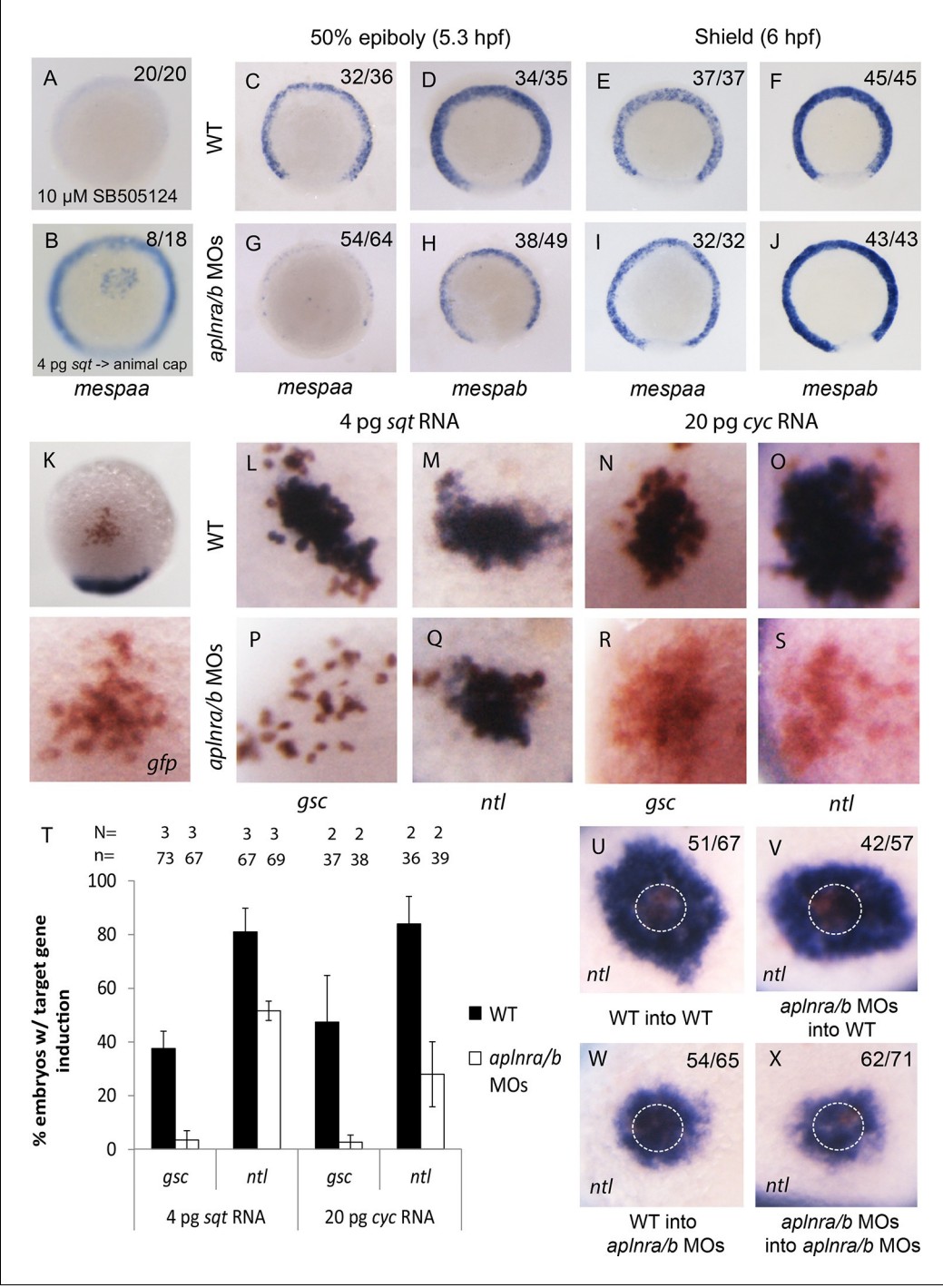

**Figure 4.** Loss of aplnr results in a delay in *mesp* gene expression and the attenuation of Sqt and Cyc activity in a non-cell autonomous manner by aplnr. (**A**) Animal view of *mespaa* expression at 50% epiboly (5.25 hpf) in embryos treated with 10 µM of SB505124 from 4–5.25 hpf. Animal cap view with dorsal to the bottom. (**B**) Animal view of *mespaa* expression at 50% epiboly (5.25 hpf) in embryos in which cells expressing 4 pg of *sqt* RNA were transplanted into the animal cap. Animal cap view with dorsal to the bottom. (**C–J**) Expression of *mespaa* and *mespab* at 50% epiboly (5.3 hpf) (**C,D,G,H**) and the shield stage (6 hpf) (**E,F,I,J**) in WT and *aplnra/b* morphant embryos when examined by WISH. Embryos are viewed from the animal pole with dorsal at the bottom. (**K**) Animal cap transplant of GFP expressing control cells detected by WISH. (**L–T**) Animal cap transplants of *sqt* or *cyc* overexpressing cells into WT (**L–O**) or *aplnra/b* morphant embryos (**P–S**) at 5.5 hpf. *gsc* and *ntl* expression is displayed in blue and *gfp* expressing donor cells are marked in brown. Both donor cells and hosts are of the same

*Figure 4 continued*

background (WT into WT or morphant into morphant). Embryos are viewed from the animal pole with dorsal at the bottom. Data are represented as means ± SEM. (**U–X**) Animal cap transplants of cells expressing high levels of *sqt* RNA at 5.5 hpf. *ntl* expression is visualized in blue and *gfp* expressing donor cells are brown. Four different combinations of donor/host cells were examined, WT into WT (donor into host) (**U**), *aplnra/b* morphant into WT (**V**), WT into *aplnra/b* morphant (**W**) and *aplnra/b* morphant into morphant (**X**). Donor cells are circled in white. Embryos are viewed from the animal pole with dorsal at the bottom.

The following source data and figure supplements are available for figure 4:

**Source data 1.** Microarray data for *mesp* family genes on *aplnra/b* morphant array.

**Figure supplement 1.** *mespaa* and *mespab* are Nodal target genes and Nodal ligand expression is not affected in *aplnra/b* morphant embryos.

**Figure supplement 2.** Aplnr is required to enhance Nodal signaling for proper cardiac development.

not increase the contribution of donor cells to the heart (22%, N=4, n=213). These experiments further argue that the heart defects observed in *aplnr* deficient embryos are suppressed if Nodal signaling is increased. To confirm that the absence of Aplnr results in lower Nodal activity, which consequently reduces/eliminates cardiogenesis, we further compromised Nodal signaling by partially depleting the embryo of Oep, the obligate Nodal co-receptor (*Gritsman et al., 1999*). In conditions where *oep* MO injections induced cardia bifida (*Figure 3O–P*), we observed that triple *aplnra/ aplnrb/oep*-depleted embryos displayed more severe defects, including cyclopia, and were completely devoid of *myl7* expression at 1 dpf (*Figure 3Q–R*). Collectively these results argue that Aplnra/b directly promotes Nodal signaling to ensure proper heart formation, with lower levels of Nodal signal being received by presumptive cardiac progenitors in the absence of Aplnr function.

## Loss of aplnr leads to a delay in *mesp* expression

We next assessed how a reduction in Nodal signaling may cause a delay in mesendodermal ingression during gastrulation. Previous work has shown that Nodal target genes become activated depending on the dose and/or associated time of exposure to the Nodal ligand (*Hagos and Dougan, 2007*). We hypothesized that Aplnr may be required to boost the Nodal signal in order to activate the expression of genes required for ingression at the right time. A particularly interesting category of genes that were down-regulated in the double *aplnra/b* morphant microarray was the *mesp* family of transcription factors (*Figure 4—source data 1*). In mice, *Mesp1/2* have been shown to regulate the migration of mesoderm through the primitive streak during gastrulation and are essential for cardiac formation (*Kitajima et al., 2000*). Treatment of 4.5–55-5.25 hpf WT embryos with Nodal inhibitor SB505124 completely abrogated *mespaa* and *mespab* expression (*Figure 4A* and *Figure 4—figure supplement 1A*). In addition, animal cap transplants of *sqt*-overexpressing cells induced *mespaa/mespab* expression, demonstrating that *mesp* genes are *bona fide* Nodal targets (*Figure 4B* and *Figure 4—figure supplement 1B*). By WISH, we confirmed that both *mespaa* and *mespab* are expressed around the margin during gastrulation (*Cutty et al., 2012*; *Sawada et al., 2000*) and that their expression is dramatically decreased in *aplnra/b* morphant embryos at 50% epiboly (5.25 hpf, *Figure 4C–D and G–H*). By carefully examining the expression pattern of *mespaa/ab* throughout development in WT and *aplnra/b* morphants, we found that at shield stage (6 hpf, 45 min after 50% epiboly) expression of both *mespaa* and *mespab* appeared to largely recover in *aplnra/b* morphant embryos (*Figure 4E–F and I–J*). Embryos at both stages were stage matched based on morphology, arguing against a general developmental delay as the cause of this phenotype. This demonstrates that Aplnr is required for timely activation of *mesp* genes and that the loss of Aplnr results in a delay in activation, rather than a general attenuation of Nodal target gene expression.

## Loss of aplnr reduces activity of Nodal ligands in a non-cell autonomous manner

We next sought to determine if Aplnr might directly act on the Nodal signaling pathway. An examination by WISH revealed that in *aplnra/b* morphants *cyc* and *sqt* expression appeared increased at 4 hpf, and normal at 4.7–5.3 hpf (*Figure 4—figure supplement 1C–N*), suggesting that attenuated Nodal signaling was not a consequence of reduced ligand expression. To address possible effects of Aplnr on Nodal signal transduction, we took advantage of a previously developed zebrafish Nodal point source assay (*Chen and Schier, 2001*). Nodal overexpressing cells were transplanted into the animal cap of host embryos and the ability to induce target genes was subsequently evaluated (*Figure 4K*). While in WT (WT host and donor) embryos 4 pg of *sqt* RNA was found to be sufficient to induce *gsc* expression, this dose was not sufficient to achieve *gsc* induction in *aplnra/b* morphant (host and donor) embryos (*Figure 4L,P and T*). To evaluate if this was a complete loss of Sqt activity or simply a reduction, we looked at the ability to induce the low threshold Nodal target gene *ntl* (*Chen and Schier, 2001*). In contrast to *gsc, ntl* expression was induced at a high frequency in both WT and morphant embryos, demonstrating that the loss of Aplnr resulted in an attenuation but not a complete loss of Sqt activity (*Figure 4M,Q and T*). Similarly, 20 pg of *cyc* RNA was less effective at inducing both *gsc* and *ntl* in *aplnra/b* morphant embryos, suggesting that Aplnra/b regulates the activity of both Nodal ligands (*Figure 4N–O, R–S and T*).

To investigate whether Aplnr is required cell autonomously for its effect on Nodal signaling a further series of transplantation experiments were conducted. For these experiments, a higher dose of *sqt* RNA (40 pg) was injected in donor embryos, which when transplanted into recipient embryos induced a ring of *ntl* expression in the host cells surrounding the donor cells (*Figure 4U*, donor cells are encircled). While both WT and *aplnra/b* morphant donor cells induced a strong ring of *ntl* expression when transplanted into WT host embryos, a significantly smaller expression domain was induced when these cells were transplanted into *aplnra/b* morphant host embryos (*Figure 4V–X*). This suggests that Aplnra/b activity is not strictly required in Nodal secreting cells for proper Nodal signaling.

## Discussion

In this study we demonstrate that the endoderm and cardiac defects in zebrafish lacking *aplnr* can be attributed to decreased Nodal signaling. We propose a model where Aplnr activity enhances the effect of Nodal signaling that allows for a Nodal threshold to be met at the right time in order to induce the expression of genes required for ingression of lateral marginal cells and proper heart formation. In the absence of *aplnr* function, a longer time frame is required for cells to reach this threshold of Nodal signaling, resulting in a delay in internalization (*Figure 4—figure supplement 2A–B*). These results are consistent with our previous observations on both the non-cell autonomous and temporal roles of Aplnr signaling in cardiac specification (*Scott et al., 2007*; *Paskaradevan and Scott, 2012*). Our conclusion on a Nodal-mediated effect of Aplnr is based on several lines of evidence in *aplnra/b* double mutants and morphants: 1) many classic Nodal target genes are downregulated; 2) ARE:Luciferase activity, a direct readout of Nodal input, is reduced; 3) sensitivity to Nodal inhibitor and lower Oep levels; 4) rescue of the phenotype by increased Nodal signal (*tar** and *lefty* MO rescue); and 5) reduced effect of Sqt and Cyc point sources.

### Aplnr: A rheostat for Nodal signaling

The link between the Aplnr defect and Nodal signaling fits well into the context of previously published literature. Nodal signaling establishes the mesendoderm and a loss of Nodal signaling or its downstream transcriptional effectors results in a heartless phenotype (*Feldman et al., 2000*; *Kunwar et al., 2003*). Furthermore, cells lacking the functional Nodal co-receptor *oep* are unable to internalize during gastrulation and cannot contribute to the mesoderm or endoderm (*Carmany-Rampey and Schier, 2001*). These cells stay at the margin and continue to move towards the vegetal pole. Likewise, *aplnra/b* morphant cells display delayed ingression kinetics that do not support proper cardiac development (*Paskaradevan and Scott, 2012*). However, unlike the loss of Oep, loss of Aplnr results in a partial and not total loss of Nodal activity, and mesendodermal ingression is evident, albeit at a later time. This also provides a plausible explanation for the incomplete penetrance

of the cardiac phenotype observed in *aplnra* and *aplnrb* mutant embryos (*Scott et al., 2007*; *Chng et al., 2013*). In mouse embryonic stem cells, graded Nodal signaling over 18 hr regulates differentiation to mesendodermal fates, with very subtle (two-fold) changes in levels of phospho-Smads having profound effects (*Lee et al., 2011*). As gastrulation proceeds far more quickly in zebrafish than it does in mice, this may also explain why profound cardiogenesis defects are not frequently seen in *Aplnr/Apj* mutant mice (*Kang et al., 2013*).

Our report of the targeted genetic knockout of zebrafish *aplnra* supports the notion that both paralogues fulfil a common role during cardiogenesis. In contrast to previous work with *aplnra* MOs (*Nornes et al., 2009*), we find that *aplnra* mutants do not have epiboly defects, but rather only share features of the *aplnrb*<sup>grinch/hu4145</sup> cardiac and endoderm phenotypes. As loss of either *aplnra* or *aplnrb* can have effects on cardiac progenitor specification, we hypothesize that both act in concert to modulate Nodal signaling, with loss of either potentially resulting in a sufficient decrease to impinge on cardiogenesis.

It has been previously demonstrated that distinct Nodal target genes require different Nodal activity thresholds for activation (*Chen and Schier, 2001*; *Lee et al., 2011*). The level of Nodal signal that a cell perceives depends on both the concentration and duration of the signal (*Hagos and Dougan, 2007*; *Dubrulle et al., 2015*). In *aplnr* mutants, given the reduction in Nodal signaling, marginal cells likely require a longer exposure to Nodal ligands before a certain threshold is reached to induce migration and *mesp* expression. This may explain why endodermal progenitors do eventually migrate and why *mesp* expression recovers in *aplnr* morphant embryos. In support of this hypothesis, *mesp* expression in *aplnr* morphant embryos at 50% epiboly appears to be retained in the most marginal blastomeres, consistent with the fact that these blastomeres are closest to the source of Nodal ligand. The basic helix-loop-helix Mesp transcription factor family has been shown to regulate the migration of mesoderm through the primitive streak in mice, and play key roles in cardiac development in several contexts (*Kitajima et al., 2000*; *Saga et al., 1999*; *Satou et al., 2004*; *Bondue et al., 2008*). This provides a molecular mechanism for how reduced levels of Nodal may translate into a delay of cell movement during gastrulation. However, we do not believe that defects in *mesp* expression can fully account for the *aplnr* cardiac phenotype. In our hands, *mespaa* overexpression was not sufficient to rescue cardiogenesis in *aplnr* morphants or mutants (A.R.D. and I.C.S., unpublished results). On the other hand, finer temporal expression of *mesp* expression may be required for proper cardiac specification.

## The nature of the aplnr effect on Nodal signaling

The cellular autonomy of Aplnr function in cardiac progenitor development has been an area of confusion, notably as both cell autonomous (*Scott et al., 2007*; *Zeng et al., 2007*) and non-autonomous (*Paskaradevan and Scott, 2012*) roles have been documented. Our results clearly show that Aplnr is not absolutely required in cells expressing Nodal ligands for signaling activity (*Figures 4U–X*), arguing against a model where Aplnr affects ligand secretion. Given that our previous experiments suggest that the Aplnr is also not required in cardiac progenitors themselves, we do not favour a model where Aplnr is required cell autonomously for reception or readout of the Nodal signal. It is possible that rather than playing a strictly cell autonomous (in cardiac progenitors) or non-autonomous (in Nodal signal sending cells) role, a threshold level of Aplnr activity is required in cells surrounding cardiac progenitors to ensure that proper levels of Nodal signaling can take place (*Figure 4—figure supplement 2C–D*). This model helps explain previous conflicting results, in which *aplnra/b* morphant donor cells typically have a reduced, but not completely absent, ability to develop as cardiomyocytes (*Scott et al., 2007*; *Paskaradevan and Scott, 2012*). It may be that in cases where a larger donor clone lacking Aplnr function is assayed, many cells inside the clone (encompassing both cardiac progenitors and other cells) do not receive the proper Nodal signal for cardiogenesis. Aplnr signaling may act, for example, to regulate Nodal ligand processing or activity, which has been shown to occur extracellularly (*Beck et al., 2002*). The interpretation of transplant experiments may therefore be confounded by the size of donor tissue. The role of a "community effect" in amplifying the Nodal signal to drive collective epithelial-to-mesenchymal transition during gastrulation has recently been described (*Voiculescu et al., 2014*). As Aplnr is both activated by Nodal (at the level of gene expression) and in turn potentiates Nodal signaling, this may provide a feed-forward mechanism to help achieve maximal Nodal signaling for proper gastrulation in a timely manner. Mechanistically, how Aplnr activity impinges on the Nodal pathway remains to be determined. Signaling

cascades downstream of Aplnr, both G protein-dependent and -independent, have been described (reviewed in (*Chapman et al., 2014*; *O'Carroll et al., 2013*). Which of these are required for Aplnr function in cardiac development, or if a new pathway is involved, remains to be elucidated. How signaling at the level of the Aplnr happens in the context of early gastrulation also remains unknown. Numerous studies have described roles for the classical Apelin/Aplnr hormone GPCR (G-protein Coupled Receptor) signaling pair in adult physiology, however in the context of early heart development Apelin does not appear to be the correct Aplnr ligand (*Scott et al., 2007*; *Chng et al., 2013*; *Ashley et al., 2005*; *Kuba et al., 2007*; *Szokodi et al., 2002*). This has been confirmed by the recent discovery of a second small endogenous peptide ligand for Aplnr, Elabela (www.elabela.com), whose mutation also results in loss of cardiac differentiation in zebrafish (*Pauli et al., 2014*; *Chng et al., 2013*) and mice (unpublished results L.H and B.R). How these two ligands may fit into the Aplnr regulation of Nodal remains an intriguing area for future investigation.

## Perspectives

In conclusion, we find that Aplnr is required to enhance Nodal signaling in order to activate genes required for proper cell movement and consequently cardiac development at the right time. This work opens several lines of future investigation on the early events required for the movement of the mesendoderm during gastrulation and early cardiac progenitor development. The levels and timing of key signaling pathways such as Nodal/TGFβ are essential to developmental output, as can be measured during differentiation of pluripotent stem cells in culture (*Kattman et al., 2011*). Similar mechanisms to that described here for Aplnr signaling may therefore remain to be discovered for other major developmental pathways. As to why lateral populations are specifically affected in *aplnra/b* mutants and not dorsal ones, where Nodal signaling is particularly prevalent, we speculate that the levels of Nodal modulated by Aplnr will not have as large a consequence in a high Nodal signaling environment like the shield/dorsal aspect of the embryo. Furthermore, only *aplnra* and not *aplnrb* is expressed in the dorsal part of the embryo (*Tucker et al., 2007*). Further, given that Aplnr signaling has been shown to regulate multiple aspects of adult physiology (reviewed in (*Chapman et al., 2014*; *O'Carroll et al., 2013*), the role of this novel signaling mechanism and the potential functions of Elabela and/or Apelin in the context of physiological homeostasis and disease (*Ho et al., 2015*; *Murza et al., 2016*) are areas of great interest.

## Materials and methods

### Zebrafish mutants, lines and imaging

Zebrafish were housed and handled as per Canadian Council on Animal Care and Hospital for Sick Children Laboratory Animal Services guidelines. Zebrafish embryos were raised at 28 degrees Celsius according to standard techniques (*Westerfield and Book, 1993*). The *Tg(myl7:EGFP)^twu34* line and *aplnrb^s608* (*grinch*, p.W90L) mutants have been previously described (*Scott et al., 2007*; *Chng et al., 2013*; *Huang et al., 2003*). *aplnrb^hu4145* (p.W54X) fish were a gift from Stefan Schulte-Merker. In the *aplnrb^hu4145* allele a STOP codon is introduced at the 54th amino acid, resulting in a severely truncated protein with no predicted function. A loss-of-function *aplnra^max* mutant line was generated with TALEN pairs purchased from ToolGen (South Korea). The TALEN-binding sites are as follows: 5' TACACCGAGACATACGATTA 3' and 5' TCACACCCAGAGTCATTATA 3'. An additional *aplnra^ins* mutant allele was purchased from Znomics, Inc. (Portland, OR). *aplnra^ZM00177433Tg* has a (c.886_887insTg(ZM)) retroviral insertion (*Amsterdam and Hopkins, 2006*) that disrupts the ORF of the single coding exon. Imaging was performed using a Leica DFC320 camera on a Leica M205FA stereomicroscope.

### Genotyping of mutants

In the *aplnrb^hu4145* mutant allele a premature stop codon has been induced into the coding sequence at amino acid position 54. Primers used for genotyping are used to amplify a 215 bp product which when cut with AciI yields fragments of 140 bp, 52 bp and 25 bp, the mutant allele will not be cut with AciI. Forward primer: CATCTTCATCCTGGGACTCACTG Reverse primer: AGCACCACATAGCTGCTGATCTT. For genotyping the allele of *aplnrb ^grinch* the same primers were used as for the *hu4145* allele, but the resultant PCR product was instead cut with EaeI, generating a 141 bp of

the 215 bp product in the mutant allele. Genotyping the *aplnra*[ZM00177433Tg] allele was performed using the following primers to detect the presence of the insertion: Forward primer: ACCC TGGAAACATCTGATGGTTC; Reverse primer: AACGGATTGAGGCAGCTGTTGAC. To determine the presence of the WT *aplnra* allele the following forward primer is used instead: Forward primer: C TCGGGTTTCTTCTGCCTTTCCT. Genotyping of the *aplnra*[max] allele was performed using the following primers to detect the absence or presence of the deletion. Forward Primer: CGCTTCAGC TTCCAGTGAG; Reverse Primer: ATGTTGACCAGCACCACGTA. To determine for the presence of the WT *aplnra* allele the following forward primer was added: Forward Primer: CACCGAGACA TACGATTACTACG. To determine for the presence of the *aplnra*[max] allele the following forward primer was added: Forward Primer: CACCGAGACATACGATTACTACTG.

## Microarrays

An Agilent zebrafish microarray (V3: 026437) was used to compare the gene expression profile of WT vs *aplnra/b* morphant embryos. 4 replicates were performed and for each experiment 20 embryos were collected at 50% epiboly and total RNA was prepared using the RNAqueous kit (Ambion, Waltham, MA). Microarray results were analyzed using Genespring v11.0.1 (Agilent Technologies, Inc., Santa Clara, CA). As recommended by the manufacturer, the data was normalized using Agilent's Spatial Detrending Lowess normalization. All data analysis was performed on log2-transformed data. Standard single factor t-tests were used followed by ranking with fold changes. After normalization and averaging the four chips, the data was filtered to remove the probes that showed no signal in order to avoid confounding effects on subsequent analysis (probes below the 20[th] percentile of the distribution of intensities were removed). MIAME-compliant microarray data was submitted to GEO (accession #GSE58683). GSEA was performed with default settings using *aplnra/b* MO-downregulated genes as a custom gene set for comparison against the *sqt* overexpression dataset from #GSE51890.

## Morpholino and RNA injection

Embryos were injected at the one cell stage according to standard procedures. Translation blocking MOs against *aplnra* (5' – cggtgtattccggcgttggctcat – 3') and *aplnrb* (5' - agagaagttgtttgtcatgtgctc – 3') have been previously described (*Scott et al., 2007*). *aplnr* morphant embryos were co-injected with 0.5 ng of *aplnrb* MOs and 1ng of *aplnra* MOs. The translation blocking MOs against *lefty1* (5' – cgcggactgaagtcatcttttcaag – 3') has been previously described (*Feldman et al., 2002*). *lefty1* morphant embryos was injected with 6 ng of MOs per embryo. Translation blocking MOs against *oep* (5' - gccaataaactccaaaacaactcga – 3') has been previously described (*Feldman and Stemple, 2001*), with 2.5 ng injected per embryo. In vitro transcribed RNA was prepared using the mMessage Machine Kit (Ambion) and purified using the MegaClear kit (Ambion). 0.5 pg of *tar\** RNA was injected per donor embryo (*Renucci et al., 1996*).

## Wholemount RNA in situ hybridization

WISH was carried out using DIG labelled antisense probes as previously described (*Thisse and Thisse, 2008*). Double ISH was performed against a fluorescein-labelled *gfp S65C* probe using previously established protocols (*Zhou et al., 2011*). Fluorescein-labelled probes were detected using INT/BCIP and DIG labelled probes with NBT/BCIP. Probes for *mespaa, mespab, sqt* and *cyc* were prepared from templates containing full length coding sequences. Probes for *myl7, nkx2.5, gsc, sox17, sox32, foxa1, foxa2, foxa3* and *flh* have been previously described (*Kikuchi et al., 2001; Schulte-Merker et al., 1994; Talbot et al., 1995; Chen and Fishman, 1996; Alexander et al., 1999; Yelon et al., 1999; Inohaya et al., 1997; Akimenko et al., 1994; Odenthal and Nüsslein-Volhard, 1998*). ImageJ analysis software was used to document *sox17* and *sox32* cell numbers and spread.

## Transplantation

Transplantation was performed as previously described (*Scott et al., 2007*). Donor embryos were injected with 5% tetramethylrhodamine dextran (10,000 MW, Molecular Probes, Waltham, MA) as a lineage tracer. Transplants were performed by placing 10–20 cells into the margin or animal cap of a

host embryo at the sphere stage (4 hpf). For *cyc/sqt* animal cap transplants 200 pg of *gfp S65C* RNA was co-injected into donor embryos. Double ISH was performed to visualize donor cells.

### Drug treatments

Nodal inhibition was performed by treating embryos with either SB505124 (10 uM) or SB431542 (Sigma, St. Louis, MO) in egg water/0.1% DMSO. The APLNR agonist ML233 was obtained from Glixx Laboratories Southborough, MA). Embryos were treated with 2.5 µM ML233 in egg water/1% DMSO from the sphere stage onwards.

### Luciferase assay

Embryos were injected with 90 pg of ARE3-luc (*Huang et al., 1995*), which contains three copies of the Activin responsive element (ARE), together with 10 pg of CMV-pRL vector (Promega, Madison, WI) at the 1-cell stage. At 30% epiboly, three groups of 20 embryos were lysed with passive lysis buffer (Promega) at room temperature for 20 min. The firefly luciferase activity, normalized to that of Renilla luciferase, was measured using the Dual luciferase assay system (Promega).

## Acknowledgements

We would like to thank all members of the Scott lab for helpful insight and discussion. Special thanks to Angela Morley for excellent zebrafish care and facility maintenance and to Jeff Burrows for generating the model figure. Thanks to Brian Ciruna, Helen McNeill and Freda Miller for helpful discussions and to Norm Rosenblum and the University of Toronto MD/PhD program for support. Thanks to Stefan Schulte-Merker for the *aplnrb*[hu4145] allele. ARD was supported by a Canadian Institutes of Health Research Vanier Canada Graduate Scholarship. This work was supported by a grant from the Canadian Institutes of Health Research to ICS (Funding Reference 123223). BR is a fellow of the Branco Weiss Foundation, an A*STAR Investigator and Young EMBO Investigator. This work is funded by a Strategic Positioning Fund on Genetic Orphan Diseases from A*STAR, Singapore.

## Additional information

### Funding

| Funder | Grant reference number | Author |
| --- | --- | --- |
| Canadian Institutes of Health Research | Operating Grant 86663 | Ashish R Deshwar<br>Ian C Scott |
| Agency for Science, Technology and Research | Strategic Positioning Fund on Genetic Orphan Diseases | Bruno Reversade |

The funders had no role in study design, data collection and interpretation, or the decision to submit the work for publication.

### Author contributions

ARD, SCC, LH, Conception and design, Acquisition of data, Analysis and interpretation of data, Drafting or revising the article; BR, ICS, Conception and design, Analysis and interpretation of data, Drafting or revising the article

### Author ORCIDs

Bruno Reversade, http://orcid.org/0000-0002-4070-7997
Ian C Scott, http://orcid.org/0000-0001-6665-1410

### Ethics

Animal experimentation: Zebrafish were housed and handled as per Canadian Council on Animal Care and Hospital for Sick Children Laboratory Animal Services (LAS) guidelines under LAS protocol number 33584.

## Additional files

### Major datasets

The following dataset was generated:

| Author(s) | Year | Dataset title | Dataset URL | Database, license, and accessibility information |
|---|---|---|---|---|
| Ashish R Deshwar, Ian C Scott | 2014 | Data From: The Apelin receptor enhances Nodal/TGFβ signaling to ensure proper cardiac development | http://www.ncbi.nlm.nih.gov/geo/query/acc.cgi?acc=GSE58683 | Publicly available at the NCBI Gene Expression Omnibus (accession no: GSE58683). |

The following previously published dataset was used:

| Author(s) | Year | Dataset title | Dataset URL | Database, license, and accessibility information |
|---|---|---|---|---|
| Nelson AC, Cutty SJ, Stemple DL, Flicek P, Bruce AE, Wardle FC | 2014 | Expression profiling of control vs. ndr1 overexpression zebrafish blastulas | http://www.ncbi.nlm.nih.gov/geo/query/acc.cgi?acc=GSE51890 | Publicly available at the NCBI Gene Expression Omnibus (accession no: GSE51890) |

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
