## [Decision Letter]

Thank you for submitting your work entitled "The Apelin receptor enhances Nodal/TGFβ signaling to ensure proper cardiac development" for consideration by *eLife*. Your article has been reviewed by three peer reviewers, and the evaluation has been overseen by a Reviewing Editor and Sean Morrison as the Senior Editor.

The reviewers have discussed the reviews with one another and the Reviewing Editor has drafted this decision to help you prepare a revised submission.

Summary:

In general all four reviewers who assessed your manuscript agree that your report represents a significant advance from your previous work in this area and revise current models about the cell autonomous/non-autonomous modes of action of the Apln receptor.

Central Conclusions:

Via the use of a novel mutant in combination with morphant analysis the authors show that Apln receptors act redundantly in specification of heart progenitors.

Apln receptor signaling is required for robust Nodal signaling, leading to an inability to efficiently up-regulate *mespaa* and *mespab* and specify cardiac progenitors.

In support of this conclusion over-expression of Nodal in the mutant/morphant embryos is sufficient to rescue the cardiac specification defects.

Essential revisions:

1) In revising the manuscript you should attempt to very clearly explain the cell autonomous versus non-autonomous roles for Aplnr in specification of the cardiac progenitors. This is somewhat confusing for the reader in light of your previous apparently contradictory findings published in Dev Cell and Biology Open. For example, wild type cells are unable to contribute to the heart in *aplnra/b* morphant host embryos, while *aplnra/b* morphant cells are able to contribute to heart in wild type embryos indicating non-autonomous role (Biol. Open 1: 275-285, 2012). However, placing *agtrl1b (aplnrb*) morphant cells in a wild-type environment did not appreciably rescue their ability to form myocardium, suggesting that Agtrl1b signaling is required autonomously in myocardial progenitors (Dev. Cell 12: 403-413, 2007). While you do discuss autonomous vs. non-autonomous roles in the Discussion, it would be helpful if your current conclusions could be summarized as a model (similar to Figure 6 of Biol. Open 2012).

2) It is also clear that there is an early endoderm defect in the double mutants. Knockdown of *aplnra* results in disrupted *sox17* expression levels and distribution, and that this is a similar phenotype to that observed in *aplnrb* knockdown. The *aplnrb* mutant has a normal endodermal organ development (Scott et al. Dev Cell), but it seems *sox17* expression at gastrulation stages has only been described in the double knockout using morpholinos, not in the single *aplnrb* mutant? Therefore, the disrupted *sox17* expression in the double knockout could simply be the consequence of knocking down the a variant. Have you analysed *sox17* expression in the *aplnrb* mutant/morphant to back up this claim?

3) Along the same lines it would be important to know if there are defects later in endodermal organ development in the double mutants. For example, does the gut develop normally, are the gut associated organs specified normally? While the cardiac phenotype (regulation of the *mesp* genes etc.) has been investigated fully, the effect of loss of *aplnr* function on the endoderm lineage, specification of which is regulated by dose-dependent Nodal signaling in both fish and mouse, was not investigated. More information should be included if possible to this point.

---

## [Author Response]

Essential revisions:

1) In revising the manuscript you should attempt to very clearly explain the cell autonomous versus non-autonomous roles for Aplnr in specification of the cardiac progenitors. This is somewhat confusing for the reader in light of your previous apparently contradictory findings published in Dev Cell and Biology Open. For example, wild type cells are unable to contribute to the heart in aplnra/b morphant host embryos, while aplnra/b morphant cells are able to contribute to heart in wild type embryos indicating non-autonomous role (Biol. Open 1: 275-285, 2012). However, placing agtrl1b(aplnrb) morphant cells in a wild-type environment did not appreciably rescue their ability to form myocardium, suggesting that Agtrl1b signaling is required autonomously in myocardial progenitors (Dev. Cell 12: 403-413, 2007). While you do discuss autonomous vs. non-autonomous roles in the Discussion, it would be helpful if your current conclusions could be summarized as a model (similar to Figure 6 of Biol. Open 2012).

We agree with the reviewers and editors that we did not fully explain our model or these published discrepancies in the original manuscript. We have altered text in the Discussion (subsection “The Nature of the Aplnr Effect on Nodal Signaling”) to address this, and based on your excellent suggestion have added new model schematics (Figure 4—figure supplement 2) to illustrate our key points. As we now articulate in the Discussion, we believe these discrepancies in the literature potentially stem from the initial size of donor clones introduced in transplants, which may influence the neighbouring environment of a cell and therefore confuse interpretations of cellular autonomy of action of the Aplnr.

2) It is also clear that there is an early endoderm defect in the double mutants. Knockdown of aplnra results in disrupted sox17 expression levels and distribution, and that this is a similar phenotype to that observed in aplnrb knockdown. The aplnrb mutant has a normal endodermal organ development (Scott et al. Dev Cell), but it seems sox17 expression at gastrulation stages has only been described in the double knockout using morpholinos, not in the single aplnrb mutant? Therefore, the disrupted sox17 expression in the double knockout could simply be the consequence of knocking down the a variant. Have you analysed sox17 expression in the aplnrb mutant/morphant to back up this claim?

We thank the editors for providing the opportunity to include a more detailed examination of endoderm defects in *aplnra, aplnrb* and *aplnra; aplnrb* mutants (see point 3 below as well). We have added images and quantification of *sox17* expressing endoderm cells during gastrulation in single *aplnrb* mutants (Figure 1, Figure 1—figure supplement 1). We find that they exhibit a decrease in both the number of expressing cells and the spread of these cells when compared to WT that is not significantly different from *aplnra* single mutants. Double *aplnra; aplnrb* mutants however exhibit a further reduction in both *sox17* cell number and spread compared to the single mutants, as would be expected if *aplnra* and *aplnrb* have partially redundant functions.

3) Along the same lines it would be important to know if there are defects later in endodermal organ development in the double mutants. For example, does the gut develop normally, are the gut associated organs specified normally? While the cardiac phenotype (regulation of the mesp genes etc.) has been investigated fully, the effect of loss of aplnr function on the endoderm lineage, specification of which is regulated by dose-dependent Nodal signaling in both fish and mouse, was not investigated. More information should be included if possible to this point.

To address this point we have performed in situ hybridization for three different markers of the endoderm (*foxa1, foxa2* and *foxa3*) in double *aplnra; aplnrb* mutants. We find that at 48 hpf endoderm formation appears defective with significant abnormalities in the most anterior pharyngeal endoderm and the formation of the liver and pancreatic buds (Figure 1—figure supplement 2). It is likely that the variability seen in these phenotypes reflects the early deficit (but not absence) of endodermal progenitors formed in double mutants, although we cannot rule out a later function for Aplnra/b in endodermal organ development.